

# Confluence does not affect the expression of miR-375 and its direct targets in rat and human insulin-secreting cell lines

Jones K. Ofori[1,2,*], Helena A. Malm[1,2,*], Ines G. Mollet[3], Lena Eliasson[1,2] and Jonathan Lou S. Esguerra[1,2]

[1] Department of Clinical Sciences Malmö, Lund University, SUS-Malmö, Sweden
[2] Lund University Diabetes Centre, Lund University, Lund and Malmö, Sweden
[3] Faculdade de Ciências Médicas, Universidade Nova de Lisboa, Lisbon, Portugal
[*] These authors contributed equally to this work.

## ABSTRACT

MicroRNAs are small non-coding RNAs, which negatively regulate the expression of target genes. They have emerged as important modulators in beta cell compensation upon increased metabolic demand, failure of which leads to reduced insulin secretion and type 2 diabetes. To elucidate the function of miRNAs in beta cells, insulin-secreting cell lines, such as the rat insulinoma INS-1 832/13 and the human EndoC-βH1, are widely used. Previous studies in the cancer field have suggested that miRNA expression is influenced by confluency of adherent cells. We therefore aimed to investigate whether one of the most enriched miRNAs in the pancreatic endocrine cells, miR-375, and two of its validated targets in mouse, *Cav1* and *Aifm1*, were differentially-expressed in cell cultures with different confluences. Additionally, we measured the expression of other miRNAs, such as miR-152, miR-130a, miR-132, miR-212 and miR-200a, with known roles in beta cell function. We did not see any significant expression changes of miR-375 nor any of the two targets, in both the rat and human beta cell lines at different confluences. Interestingly, among the other miRNAs measured, the expression of miR-132 and miR-212 positively correlated with confluence, but only in the INS-1 832/13 cells. Our results show that the expression of miR-375 and other miRNAs with known roles in beta cell function is independent of, or at least minimally influenced by the density of proliferating adherent cells, especially within the confluence range optimal for functional assays to elucidate miRNA-dependent regulatory mechanisms in the beta cell.

## INTRODUCTION

Type-2 diabetes (T2D) is a complex metabolic disease characterized by elevated blood glucose levels due to a combination of insulin resistance and impaired insulin secretion (*Prasad & Groop, 2015*). Western life-style with reduced exercise and unhealthy food-habits result in insulin resistance in target tissues such as in liver, muscle and adipose tissue. To cope with increased metabolic demands, pancreatic beta cells secrete more insulin. Failure to compensate contributes to the development of T2D (*Halban et al., 2014*). By regulating

Corresponding authors
Lena Eliasson,
lena.Eliasson@med.lu.se
Jonathan Lou S. Esguerra,
jonathan.esguerra@med.lu.se

various cellular processes within the beta cell, microRNAs (miRNAs) have been suggested to play important roles in rapid compensatory response to changing environments (*Eliasson & Esguerra, 2014*; *Esguerra et al., 2014*).

MiRNAs are small non-coding RNAs involved in the regulation of gene expression. They bind to the 3'UTR of the target mRNA leading to mRNA degradation and/or translational repression (*Bartel, 2009*). In diabetes, several miRNAs have been shown to be differentially expressed and have been shown to be involved in important beta cell functions and in maintaining beta cell mass (*Esguerra et al., 2014*; *Poy et al., 2009*).

The first miRNA discovered in the pancreatic islet cells was miR-375 (*Poy et al., 2004*), which is one of the most highly-enriched miRNAs in the pancreatic islets. Since its discovery, miR-375 has been shown to negatively regulate a plethora of genes involved in pancreatic beta cell function (*Eliasson, 2017*) such as in insulin secretion by regulating myotrophin (*Mtpn*) (*Poy et al., 2004*) and various voltage-gated sodium channels (SCNs) (*Salunkhe et al., 2015*). Knock out of miR-375 in mouse (*375KO*), resulted in hyperglycaemic animals with defective proliferative capacity of endocrine cells leading to decreased beta cell mass (*Poy et al., 2009*). Studies on islets of *375KO* mice reveal direct regulation of multiple genes involved in the negative control of cellular growth and proliferation such as the apoptosis-inducing factor, mitochondrion-associated 1 (*Aifm1*) and caveolin1 (*Cav1*) (*Poy et al., 2009*).

Another highly-enriched beta cell miRNA is miR-200a, demonstrated to be upregulated in islets of the db/db diabetic mouse model and shown to contribute in regulating pancreatic beta cell survival in T2D (*Belgardt et al., 2015*). There are also a number of miRNAs such as miR-132, miR-212, miR-130a and miR-152 shown to be upregulated in the pancreatic islets of the widely-studied T2D model Goto-Kakizaki rats (*Esguerra et al., 2011*) with active roles in beta cell stimulus-secretion coupling (*Malm et al., 2016*; *Ofori et al., 2017*).

Cell lines are commonly utilized to unravel the molecular mechanisms by which miRNAs participate in cellular processes. The ease of handling, maintenance and availability of cell line models make them indispensable tools in molecular biology investigations. Indeed, studying molecular mechanisms underlying fundamental beta cell processes such as the stimulus-secretion coupling have been made possible by tumour-derived rat insulin-secreting cell lines such as INS-1 (*Asfari et al., 1992*), and its more recent derivative sub-line, INS-1 832/13 cells (*Hohmeier et al., 2000*). Recently, the human beta cell line EndoC-βH1 has also been made available which further enabled deeper investigations of molecular mechanisms governing insulin secretion in humans (*Andersson et al., 2015*; *Ravassard et al., 2011*).

It has been noticed that the global abundance of miRNAs is generally lower in cell lines than in primary tissues of cancer (*Lu et al., 2005*). One hypothesis is that the tight cell–cell contacts in primary tissues contribute to the activation of miRNA biogenesis. Indeed, it was shown that higher cellular density or confluence resulted in higher levels of various miRNAs in HeLa and NIH3T3 cells (*Hwang, Wentzel & Mendell, 2009*; *Van Rooij, 2011*). Because many functional assays, e.g., insulin secretion assay, being performed on beta cell lines require optimal culture conditions including cell densities, we therefore set out to investigate whether confluence affects the expression of miR-375 and two of its

validated targets in the mouse beta cell, *Aifm1* and *Cav1*, in the rat INS-1 832/13 cells and in the human EndoC-βH1 cells. We also investigated the influence of confluence on the expression levels of miR-200a, miR-130a, miR-152, miR-132 and miR-212.

## MATERIALS & METHODS

### Reagents

All reagents were purchased from Sigma Aldrich (St. Louis, MO, USA) unless otherwise stated.

### Cell culture, seeding and imaging

EndoC-βH1 cells (EndoCells, Paris, France) (*Andersson et al., 2015*; *Ravassard et al., 2011*) (passages between 76–80) were seeded on 24-well plates coated with Matrigel-fibronectin (100 μg/mL and 2 μg/mL; Sigma-Aldrich, Steinheim, Germany) at the following densities: 390,000 cells/well, 312,000 cells/well, 234,000 cells/well and 78,000 cells/well to reach 100%, 80%, 60% and 20% estimated confluence respectively after 48 h. The cells were maintained in a culture medium containing: DMEM (5.6 mM glucose), 2% BSA fraction V (Roche Diagnostics, Mannheim, Germany), 10 mM nicotinamide (Merck Millipore, Darmstadt, Germany), 50 μM 2-mercaptoethanol, 5.5 μg/mL transferrin, 6.7 ng/mL sodium selenite (Sigma-Aldrich), 100 U/mL penicillin, and 100 μg/mL streptomycin (PAA Laboratories, Pasching, Austria).

Rat insulinoma INS-1 832/13 cells (passages between 50–55) (*Hohmeier et al., 2000*) were seeded accordingly: 300,000 cells/well, 240,000 cells/well, 180,000 cells/well and 60,000 cells/well to reach 100%, 80%, 60% and 20% estimated confluence respectively after 48 h. The growth area per well in the 24 well plate is 1.9 cm$^2$. Cells were maintained in RPMI 1640 medium containing 11.1 mM glucose (HyClone, UT, USA) as previously described (*Salunkhe et al., 2015*). Both cell lines were incubated in a humidified atmosphere with 5% $CO_2$ at 37 °C for 48 h. All experiments were performed within culture passages in which the cell lines respond robustly to glucose-stimulated insulin secretion assay.

To measure cell-to-cell contact, 300 uL suspension of EndoC-βH1 cells were seeded on microscope slides with 8 chambered wells (growth area per well: 1.0 cm$^2$) (Cat. No. 80827; ibidi Gmbh, Planegg, Germany). The corresponding seeding cell densities for each estimated harvest confluency were as follows: 20% confluence: 41,053 cells/cm$^2$; 60% confluence: 123,158 cells/cm$^2$; 80% confluence: 1642,10 cells/cm$^2$; 100% confluence: 205,263 cells/cm$^2$. The seeded cells were imaged at 40× magnification on a Zeiss LSM 510 microscope. Cell-to-cell distances between randomly selected cells were measured ($n = 40$–90) and averaged. See Fig. S1 for representative image panels with distance measurements.

### RNA extraction, RT-PCR and qPCR

RNA extraction, RT-PCR for total RNA and stem-loop RT-qPCR for microRNA was performed as previously described (*Salunkhe et al., 2015*) using the following TaqMan® miRNA Assays: miR-375 (RT_000564), miR-200a (RT_000502), miR-130a (RT_000454), miR-152 (RT_000475), miR-132 (RT_000457), miR-212 (RT_002551), rat U87

(RT_001712), human RNU44 (RT_001094) and human RNU48 (RT_001006) for generating cDNA. The following primers from TaqMan® Gene Expression and TaqMan® miRNA Assays were used for qPCR: Cav1/CAV1 (Rn00755834_m1/Hs00971716_m1), Aifm1/AIFM1 (Rn00442540_m1/ Hs00377585_m1), miR-375 (TM_ 000564), miR-200a (TM_000502), miR-130a (TM_00454), miR-152 (TM_000475), miR-132 (TM_000457) and miR-212 (TM_002551) were used for qPCR. Hprt1/HPRT1 (Rn_01527840/4333768F) and Ppia/PPIA (Rn_ 00690933/4333763F) were used as endogenous controls for mRNA expression, while rat U87 (TM_001712) or human RNU44 (TM_001094) and human RNU48 (TM_001006) were used as endogenous control for miRNA expression. The relative quantities were calculated using the $\Delta\Delta C_t$ method. The average $C_t$ values of qPCR assays for each duplicate or triplicate runs are provided in Table S1.

### Statistical analysis

Differences between groups were tested using one-way ANOVA followed by Tukey's multiple comparison test as implemented in GraphPad Prism 7. Data are presented as mean ± SEM.

## RESULTS & DISCUSSION

To find out the influence of cell confluence on the expression of selected miRNAs and targets, we utilized the rat (INS-1 832/13) and human (EndoC-βH1) insulin-secreting cell lines seeded at different cell densities, followed by gene expression measurements (Fig. 1A). To quantify the cell-to-cell contact, we also seeded EndoC-βH1 cells in parallel, at different densities corresponding to each confluence harvest point. We measured on average 30 µm between cells in 100% confluent plates, while the lowest 20% confluent plates contained cells with an average of 140 µm cell-to-cell distance (Fig. 1B and Fig. S1).

This study mainly addressed the issue whether confluence affects miR-375 expression, as it is one of the most enriched miRNAs in the pancreatic beta cells influencing diverse molecular processes, from insulin secretion to cellular growth and proliferation (*Eliasson, 2017*; *Poy et al., 2004*; *Poy et al., 2009*; *Salunkhe et al., 2015*). The genes *Aifm1* and *Cav1* are among the many genes shown to be directly targeted by miR-375 in mouse beta cells. The negative effect of miR-375 on both the mRNA and protein levels of the two genes has been demonstrated, and in the islets of 375KO mice, increased expression of these targets was also detected at the mRNA level (*Poy et al., 2009*). *Aifm1* and *Cav1* are involved in signaling mechanisms that negatively regulate cellular growth and proliferation, hence *375KO* mice were found to have reduced beta cell mass and defective proliferative capacity in the pancreatic endocrine cells (*Poy et al., 2009*).

In the rat insulin-secreting cell line, INS-1 832/13, we previously showed the reduction of *Aifm1* and *Cav1* mRNA expression upon miR-375 over-expression delineating the conserved targeting in rodents of these genes by miR-375 (*Salunkhe et al., 2015*). In humans, computational predictions show that miR-375 has two non-conserved target sites in the 3'UTR of *AIFM1* mRNA (Target Scan v.7.1 release June 2016) (*Agarwal et al., 2015*), and one target site in the *CAV1* 3'UTR identified by another miRNA target prediction program (RNA22 algorithm implemented at miRWalk 2.0) (*Dweep & Gretz, 2015*).

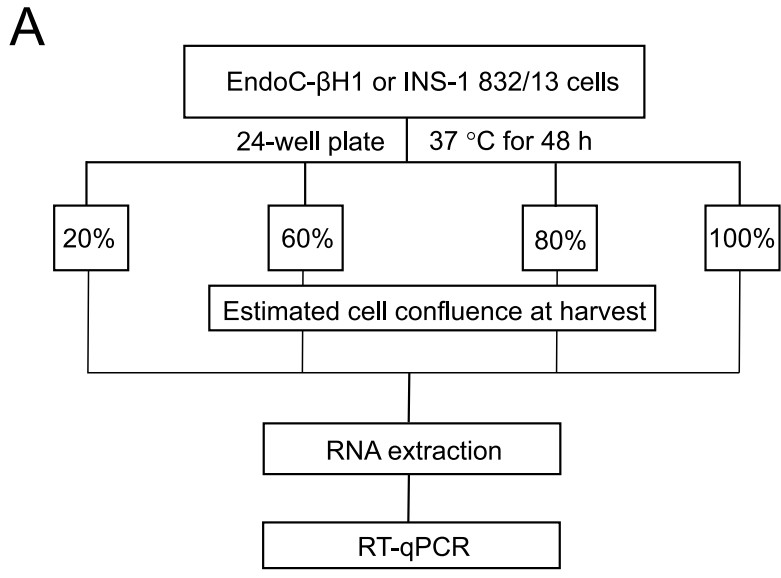

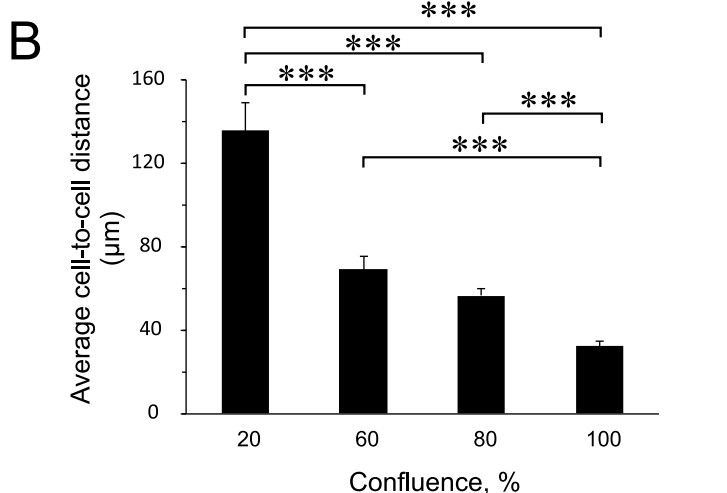

**Figure 1   Experimental design and average cell-to-cell distance at harvest.** (A) Rat (INS-1832/13) and human (EndoC-βH1) insulin-secreting cell lines were seeded at different cell densities prior to downstream assays as outlined. (B) The average distance between cells at different harvest confluences for EndoC-βH1 cells. Data are presented as mean $\pm$ SEM of $N = 40$–90 distance measurements from three independent seedings. ($\ast\ast\ast$) $p < 0.001$; one-way ANOVA Tukey's multiple comparison test.

In INS-1 832/13 cells, we did not detect any significantly altered expression of neither miR-375 nor its targets among the different confluences (Figs. 2A–2C). Likewise in the human EndoC-βH1 cells, the expression of miR-375 was similar at all confluences (Fig. 3D). Interestingly, we observed slightly decreased expression of *CAV1* mRNA at 60% and 100% confluence compared to the 20% confluence (Fig. 3F). Although miR-375 is also predicted to target *CAV1* 3'UTR mRNA, miR-375 expression was not elevated at higher confluences, implying that *CAV1* mRNA is potentially regulated by other factors. These results further

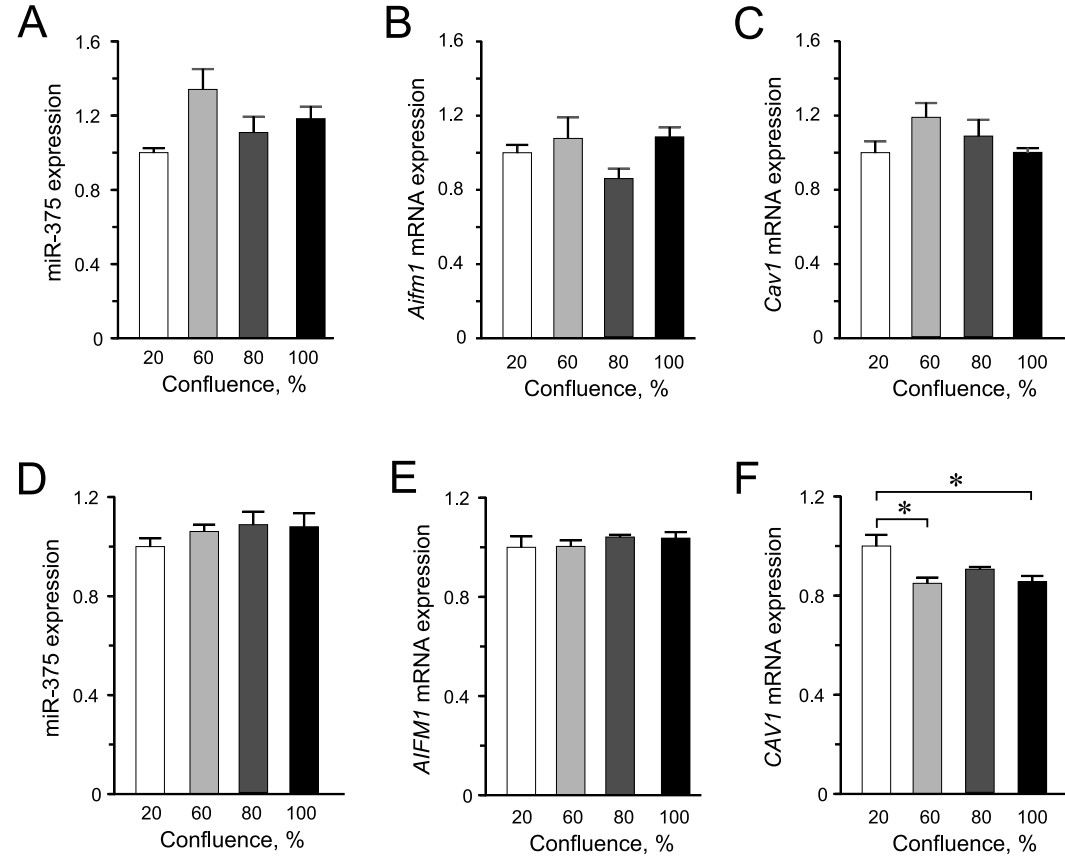

**Figure 2** **Expression of miR-375 and its targets in INS-1 832/13 cells (A–C) or in EndoC-βH1 cells (D–F).** (A) miR-375 expression at different cell confluence of INS-1 832/13 cells. Expression was normalized to rat *U87*. (B and C) *Aifm1* and *Cav1* expression in INS-1 832/13 cells, respectively. Expression was normalized to *Hprt1* and *Ppia*. (D) Expression of miR-375 at different cell density in EndoC-βH1 cells. Expression was normalized to human *RNU44* and *RNU48*. (E and F) *AIFM1* and *CAV1* expression in EndoC-βH1 cells, respectively. Expression were normalized to *HPRT1* and *PPIA*. For all experiments, data are presented as mean of $N = 3$–4 biological replicates, (∗) $p < 0.05$ using one-way ANOVA Tukey's multiple comparison test.

underline the impact of species-specific effects of miRNA-mediated regulation in cellular processes.

Among the other miRNAs included in this study, we observed significantly higher expression levels of miR-132 and miR-212 at higher confluences in INS-1 832/13 cells (Figs. 3A–3B) but only an increasing trend in the human EndoC-βH1 cells (Figs. 3C–3D). For miR-200a, miR-130a and miR-152, the expression levels were found not to be influenced by cellular confluence (Fig. S2). However, although not significant, we observed a trend of increasing miRNA expression from 20% to higher confluences in the EndoC-βH1 cells (Figs. S2D–S2F). Overall, the pattern of increased miRNA expression with increasing confluence in this study, supports previous observation of increased activation of miRNA biogenesis and expression at higher cellular densities (*Hwang, Wentzel & Mendell, 2009*; *Van Rooij, 2011*).

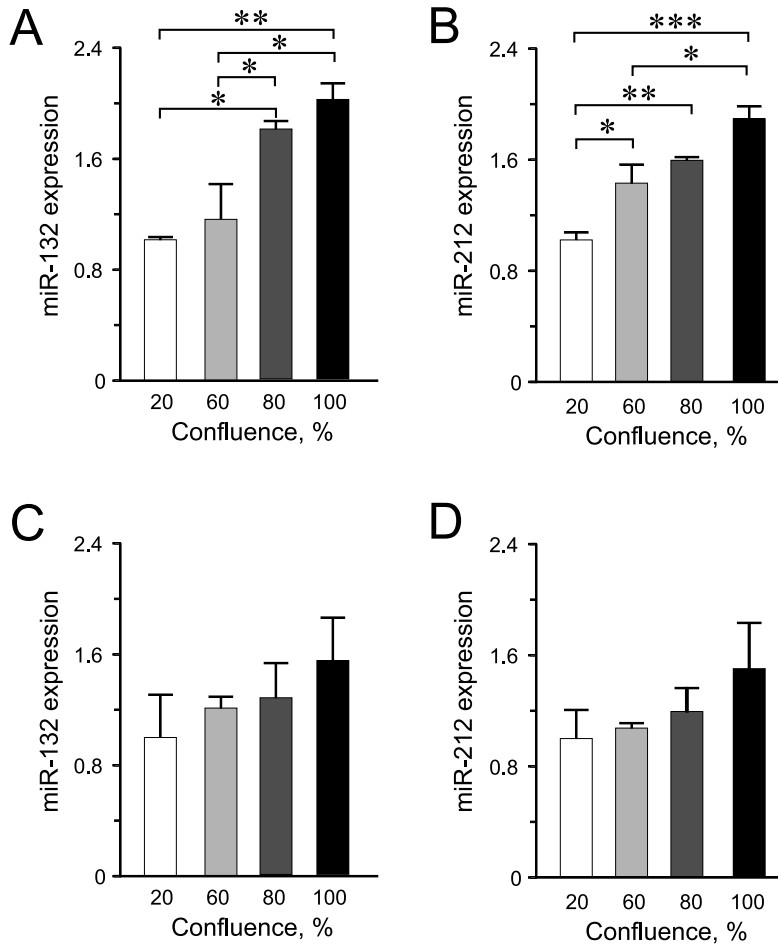

**Figure 3** **miR-132 and miR-212 expression in INS-1 832/13 cells (A–B) or in EndoC-βH1 cells (C–D) at different confluences.** Expression was normalized to rat *U87* or to human *RNU44* and *RNU48*. Data are presented as mean ± SEM of *N* = 3–4 biological replicates. (∗) *p* < 0.05, (∗∗) *p* < 0.01, (∗∗∗) *p* < 0.001; one-way ANOVA Tukey's multiple comparison test.

## CONCLUSION

We found virtually no significant differences in the expression levels of miR-375, *CAV1* mRNA and *AIFM1* mRNA at higher confluences, from 60%–100%, either in the rat or human beta cell lines. Moreover, we did not find significant differences in the expression of the other miRNAs tested in either INS-1 832/13 or EndoC-βH1 cells between 80% and 100% confluence. These results are comforting because most functional assays employing pancreatic beta cell lines utilize these confluence levels to attain consistent results. For instance, to ensure optimal insulin secretion in cultured beta cell lines, insulin-secretion assays are commonly performed when the cell culture confluence is at least 90%.

Although we showed that miR-375, which is one of the most enriched beta cell miRNA was not significantly influenced by confluence level in cultured rat and human beta cell lines, we clearly demonstrated that miR-132 and miR-212 are more dependent on cellular densities, as was shown for some miRNAs in other cells types (*Hwang, Wentzel & Mendell,*

*2009*; *Van Rooij, 2011*). One must therefore be cautious in controlling for cell densities when investigating specific miRNAs in *in vitro* systems.

It has been observed that primary tissues generally exhibit higher global miRNA abundance compared to cell lines in part attributed to tighter, and greater cell-to-cell contact in three-dimensions (*Lu et al., 2005*). Nevertheless, it remains to be seen in the pancreatic endocrine cells how the three-dimensional organization of the cells impacts the global miRNA expression and hence, the regulation of various cellular processes.

## ACKNOWLEDGEMENTS

We thank Britt-Marie Nilsson and Anna Maria V. Ramsay for the technical support. Special acknowledgment to R Scharfmann and P Ravassard, INSERM and ENDOCELL for providing us with the EndoC-βH1 human beta cell line, and C Newgard and H Mulder for providing us with the INS-1 832/13 cells.

### Funding

This work was supported by Swedish Research Council (LE; 2016-02124), Linnaeus grant to LUDC, SFO-EXODIAB, Region Skåne ALF, Swedish Foundation for Strategic Research-IRC-LUDC, The Swedish diabetes foundation (LE), Albert Påhlsson Foundation, Region Skåne ALF (LE), Magnus Bergvalls Stiftelse (JLSE), The Edla and Eric Smedberg Foundation Fund through The Royal Physiographic Society of Lund, and Crafoord Foundation (JLSE). The funders had no role in study design, data collection and analysis, decision to publish, or preparation of the manuscript.

### Grant Disclosures

The following grant information was disclosed by the authors:
Swedish Research Council: LE; 2016-02124.
Linnaeus grant to LUDC; SFO-EXODIAB.
Swedish Foundation for Strategic Research - IRC LUDC.
Region Skåne ALF (LE) Boehringer-Ingellheim.
Swedish Diabetes Foundation (LE).
Crafoord Foundation (JLSE).
Albert Påhlsson Foundation.
The Edla and Eric Smedberg Foundation Fund through The Royal Physiographic Society of Lund.
Magnus Bergvalls Stiftelse (JLSE).

### Competing Interests

The authors declare there are no competing interests.

## Author Contributions

- Jones K. Ofori, Helena A. Malm and Ines G. Mollet conceived and designed the experiments, performed the experiments, analyzed the data, wrote the paper, prepared figures and/or tables, reviewed drafts of the paper.
- Lena Eliasson conceived and designed the experiments, analyzed the data, contributed reagents/materials/analysis tools, wrote the paper, prepared figures and/or tables, reviewed drafts of the paper.
- Jonathan Lou S. Esguerra conceived and designed the experiments, performed the experiments, analyzed the data, contributed reagents/materials/analysis tools, wrote the paper, prepared figures and/or tables, reviewed drafts of the paper.

## Data Availability

The raw data has been supplied as a Supplementary File.

## Supplemental Information

Supplemental information for this article can be found online at http://dx.doi.org/10.7717/peerj.3503#supplemental-information.

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
