# Peer review of "Confluence does not affect the expression of miR-375 and its direct targets in rat and human insulin-secreting cell lines"

_PeerJ, doi:10.7717/peerj.3503_

## Round 0.1 · original submission · Major Revisions

Dear Dr. Esguerra,

Although two of the reviewers have given very favorable comments, reviewer 2 has risen very good concerns, which need to be addressed. I will look forward for your rebuttal and addressing all the points .

·

Basic reporting

No comment

Experimental design

Please provide a reference or supporting data justifying the initial cells/well (line 104, 105, 112,113) that were seeded to obtain the necessary confluence in 48 hours.

Validity of the findings

Please provide p values for the ANOVA Tukey test for Figure 2A, 2B

Reviewer 2 ·

Basic reporting

1. Clear and unambiguous, professional English used throughout: The manuscript has a lot of grammatical errors. Commas after sentences are missing in the manuscript. The manuscript has to edited for grammatical mistakes.
2. Literature references, sufficient field background/context provided: No Comment
3. Professional article structure, figs, tables. Raw data shared: No Comment
4. Self-contained with relevant results to hypotheses: The results do not completely justify the hypotheses as additional experimental results are necessary to support the hypotheses.

Experimental design

1. Original primary research within Aims and Scope of the journal: No comment
2. Research question well defined, relevant & meaningful. It is stated how research fills an identified knowledge gap: No comment
3. Rigorous investigation performed to a high technical & ethical standard: No comment
4. Methods described with sufficient detail & information to replicate: The manuscript describes the use of two cell lines EndoC-βH1 and Rat Insulinoma INS-1832/13 cells. But it is imperative to mention the passage numbers of cells used for the experiments for replication by other investigators. Further, it is necessary to mention the effect of passage number on the outcome of the results.

Validity of the findings

1. Impact and novelty not assessed. Negative/inconclusive results accepted. Meaningful replication encouraged where rationale & benefit to literature is clearly stated: No comment
2. Data is robust, statistically sound, & controlled: No comment
3. Conclusion are well stated, linked to original research question & limited to supporting results: No comment
4. Speculation is welcome, but should be identified as such: No comment

Additional comments

The manuscript clearly shows the cell confluency has no effect on the expression of mir-375 in both rat and human insulin-secreting cell lines. Although the results are promising, there are significant drawbacks which need to be addressed before it can be published. The following points need to be addressed before it can be concluded that there are no significant differences in the expression levels of mir-375, CAV1 mRNA, and AIFM1 mRNA at higher confluences in rat or human beta cell lines.
1. In the introduction section-Line 97, it has been mentioned targets CAV1 mRNA and AIFM1 mRNA are from mouse beta cell line, but the authors have used rat and human beta cell line for experiments. This needs to be corrected as it is confusing.
2. In the methods section, the cell passage numbers have to be mentioned for replication of experimental procedures and also the effect of cell passage on the expression levels of mir-375, CAV1 mRNA, and AIFM1 mRNA has to be determined. This is vital as most of the cell lines lose insulin secreting ability with passages.
3. The experiments have been done under normal glucose conditions. So the expression levels of mir-375, CAV1 mRNA and AIFM1 mRNA has to be defined under hyperglycemic conditions as glucose homeostasis is affected during diabetic conditions.
4. Authors in their previous study have demonstrated that ectopic expression of mir-375 downregulates Aifm 1 and Cav 1 mRNA expression in rat insulin-secreting cell lines. But they also mention that by computational predictions show that miR-375 has no target site on Cav 1 mRNA 3’ UTR. It would be much more convincing if the authors can show that effect of ectopic expression of mir-375 on Aifm 1 and Cav 1 mRNA expression in the human insulin-secreting cell line. Also, why did the authors choose Cav 1 mRNA for their experiments with human cell lines when miR-375 cannot regulate its expression?
5. Authors show only the mRNA expression of CAV1 and AIFM1. It would be more convincing if they show the protein expression of these genes. Also, quantification of mRNA levels of these genes by performing Northern blot analysis would validate the qPCR results.
6. The names of CAV 1 and AIFM 1 should be uniform throughout the manuscript as in line 145 of the results and discussion section, it is in capital letters, and in the rest of the manuscript, it is in lower case.

·

Basic reporting

No comment

Experimental design

No comment

Validity of the findings

No comment

Additional comments

Article is well written. It would be really interesting to see what happens with other miRNAs.There are really great pool of miRNAs in the beta cells. It would have been nice if you included 2 or more miRNAs for comparison.

The research paper is crisp and well written. In the introduction they have used sufficient references to put forth their idea for investigation. And not found any grammatical and typographical mistakes.

The experimental designs were simple and accurate. The important idea of the paper is talking about whether the confluency of the cells affects the expression of an important micro RNA. In general, for experimental procedures researchers use 60 to 80% confluency of the cells and it’s been seen, in some cases confluency affects the cell behavior and gene expressions.

The outcome of the paper shows there were no significant expression changes of miR 375 and the two target genes sufficiently well supported by the experiments and the data they provided. But it would have been interesting to see how some other miRNAs would have behaved if they have included one or more of the miRNAs in the experimental design.

The reason I recommend this paper for acceptance is the main idea, that the confluency does not change any difference in the expression. This would give cell biologists hope while designing experiments. Though we should still keep in mind that these results could change with the type of cell lines and type of the targets used.

---

## Round 0.2 · accepted · Accept

Thank you for revising the manuscript as suggested by the reviewers and for your rebuttal. I believe you have addressed the concerns raised by the reviewers efficiently, that I can accept this manuscript and recommend for publication in PeerJ. Congratulations!

·

Basic reporting

NONE

Experimental design

NONE

Validity of the findings

NONE

Additional comments

The authors satisfactorily addressed the concerns that were raised in the first round of review.